# Genomic islands targeting *dusA* in *Vibrio* species are distantly related to *Salmonella* Genomic Island 1 and mobilizable by IncC conjugative plasmids

Romain Durand🄳¤, Florence Deschênes🄳, Vincent Burrus🄳*

Département de biologie, Université de Sherbrooke, Sherbrooke, Québec, Canada

¤ Current address: Institut de Biologie Intégrative et des Systèmes, Université Laval, Québec, Canada
* vincent.burrus@usherbrooke.ca

**Data Availability Statement:** All relevant data are within the manuscript and its Supporting Information files.

## Abstract

*Salmonella* Genomic Island 1 (SGI1) and its variants are significant contributors to the spread of antibiotic resistance among *Gammaproteobacteria*. All known SGI1 variants integrate at the 3' end of *trmE*, a gene coding for a tRNA modification enzyme. SGI1 variants are mobilized specifically by conjugative plasmids of the incompatibility groups A and C (IncA and IncC). Using a comparative genomics approach based on genes conserved among members of the SGI1 group, we identified diverse integrative elements distantly related to SGI1 in several species of *Vibrio*, *Aeromonas*, *Salmonella*, *Pokkaliibacter*, and *Escherichia*. Unlike SGI1, these elements target two alternative chromosomal loci, the 5' end of *dusA* and the 3' end of *yicC*. Although they share many features with SGI1, they lack antibiotic resistance genes and carry alternative integration/excision modules. Functional characterization of IME*Vch*USA3, a *dusA*-specific integrative element, revealed promoters that respond to AcaCD, the master activator of IncC plasmid transfer genes. Quantitative PCR and mating assays confirmed that IME*Vch*USA3 excises from the chromosome and is mobilized by an IncC helper plasmid from *Vibrio cholerae* to *Escherichia coli*. IME*Vch*USA3 encodes the AcaC homolog SgaC that associates with AcaD to form a hybrid activator complex AcaD/SgaC essential for its excision and mobilization. We identified the *dusA*-specific recombination directionality factor RdfN required for the integrase-mediated excision of *dusA*-specific elements from the chromosome. Like *xis* in SGI1, *rdfN* is under the control of an AcaCD-responsive promoter. Although the integration of IME*Vch*USA3 disrupts *dusA*, it provides a new promoter sequence and restores the reading frame of *dusA* for proper expression of the tRNA-dihydrouridine synthase A. Phylogenetic analysis of the conserved proteins encoded by SGI1-like elements targeting *dusA*, *yicC*, and *trmE* gives a fresh perspective on the possible origin of SGI1 and its variants.

**Funding:** This work was supported by Discovery Grants (RGPIN-2016-04365 and RGPIN-2021-02814) from the Natural Sciences and Engineering Council of Canada (NSERC, https://www.nserc-crsng.gc.ca/) and Project Grant (PJT-153071) from the Canadian Institutes of Health Research (CIHR, https://cihr-irsc.gc.ca/e/) to V.B. R.D. is the recipient of a Fonds de recherche du Québec-Nature et Technologies (FRQNT, https://frq.gouv.qc.ca/) doctoral fellowship. The funders had no role in study design, data collection and analysis, decision to publish, or preparation of the manuscript.

## Author summary

We identified integrative elements distantly related to *Salmonella* Genomic Island 1 (SGI1), a key vector of antibiotic resistance genes in *Gammaproteobacteria*. SGI1 and its variants reside at the 3' end of *trmE*, share a large, highly conserved core of genes, and carry a complex integron that confers multidrug resistance phenotypes to their hosts. Unlike members of the SGI1 group, these novel genomic islands target the 5' end *dusA* or the 3' end of *yicC*, lack multidrug resistance genes, and seem much more diverse. We showed here that, like SGI1, these elements are mobilized by conjugative plasmids of the IncC group. Based on comparative genomics and functional analyses, we propose a hypothetical model of the evolution of SGI1 and its siblings from the progenitor of IncA and IncC conjugative plasmids via an intermediate *dusA*-specific integrative element through gene losses and gain of alternative integration/excision modules.

## Introduction

Integrative and mobilizable elements (IMEs) are discrete, mobile chromosomal regions that can excise from the chromosome and borrow the mating apparatus of helper conjugative elements to transfer to a new bacterial host [1,2]. IMEs are usually composed of two main functional modules. The site-specific recombination module contains genes and *cis*-acting sequences that mediate the integration of the IMEs into and their excision from the chromosome. The mobilization module includes the *cis*-acting origin of transfer (*oriT*) and usually encodes mobilization proteins required to initiate the conjugative transfer at *oriT* [1]. In its simplest form, the mobilization module only consists of an *oriT* locus mimicking the *oriT* of the helper element [3–5]. The excision of IMEs is elicited by conjugative plasmids or integrative and conjugative elements (ICEs). These helper elements encode the type IV secretion system (T4SS) that translocates the IME DNA into the recipient cell [1].

Several distinct families of IMEs have been described to date. Most encode beneficial traits for their host, such as resistance to antibiotics and heavy metals or bacteriocin synthesis [1,6]. *Salmonella* Genomic Island 1 (SGI1) is certainly one of the most studied IMEs. Though first described 20 years ago, SGI1 and its siblings have only recently gained a lot of attention due to their prevalence and prominent role in the spread of multidrug resistance [7,8]. The canonical 43-kb SGI1 resides at the 3' end of *trmE* (also known as *mnmE* or *thdF*) in *Salmonella enterica* serovar Typhimurium DT104 [9]. *trmE* encodes the 5-carboxymethylaminomethyluridine-tRNA synthase GTPase subunit. SGI1 variants have been reported in a wide array of *Gammaproteobacteria*, including *Proteus mirabilis* (PGI1), *Acinetobacter baumannii* (AGI1), *Morganella*, *Providencia*, *Enterobacter*, *Escherichia coli*, *Vibrio cholerae* (GI-15), and *Klebsiella pneumoniae* [7,10,11]. Most variants carry a class I integron structurally similar to the In104 integron of SGI1. In104 confers resistance to ampicillin, chloramphenicol/florfenicol, streptomycin/spectinomycin, sulfamethoxazole, and tetracycline [8,12]. SGI1 and its variants are an epidemiological threat exacerbated by their specific mobilization by conjugative plasmids of the incompatibility groups A (IncA) and C (IncC) [13,14]. IncC plasmids contribute to the global circulation of multidrug resistance genes, including NDM metallo-β-lactamase and carbapenemase genes, among a broad range of *Gammaproteobacteria* [15,16]. The transcriptional activator AcaCD encoded by IncC plasmids triggers the excision and mobilization of SGI1 [17,18].

SGI1 and most variants share a conserved core set of 28 genes, representing 27.4 kb, disrupted by insertion sequences and the class 1 integron inserted at diverse positions (Fig 1, top)

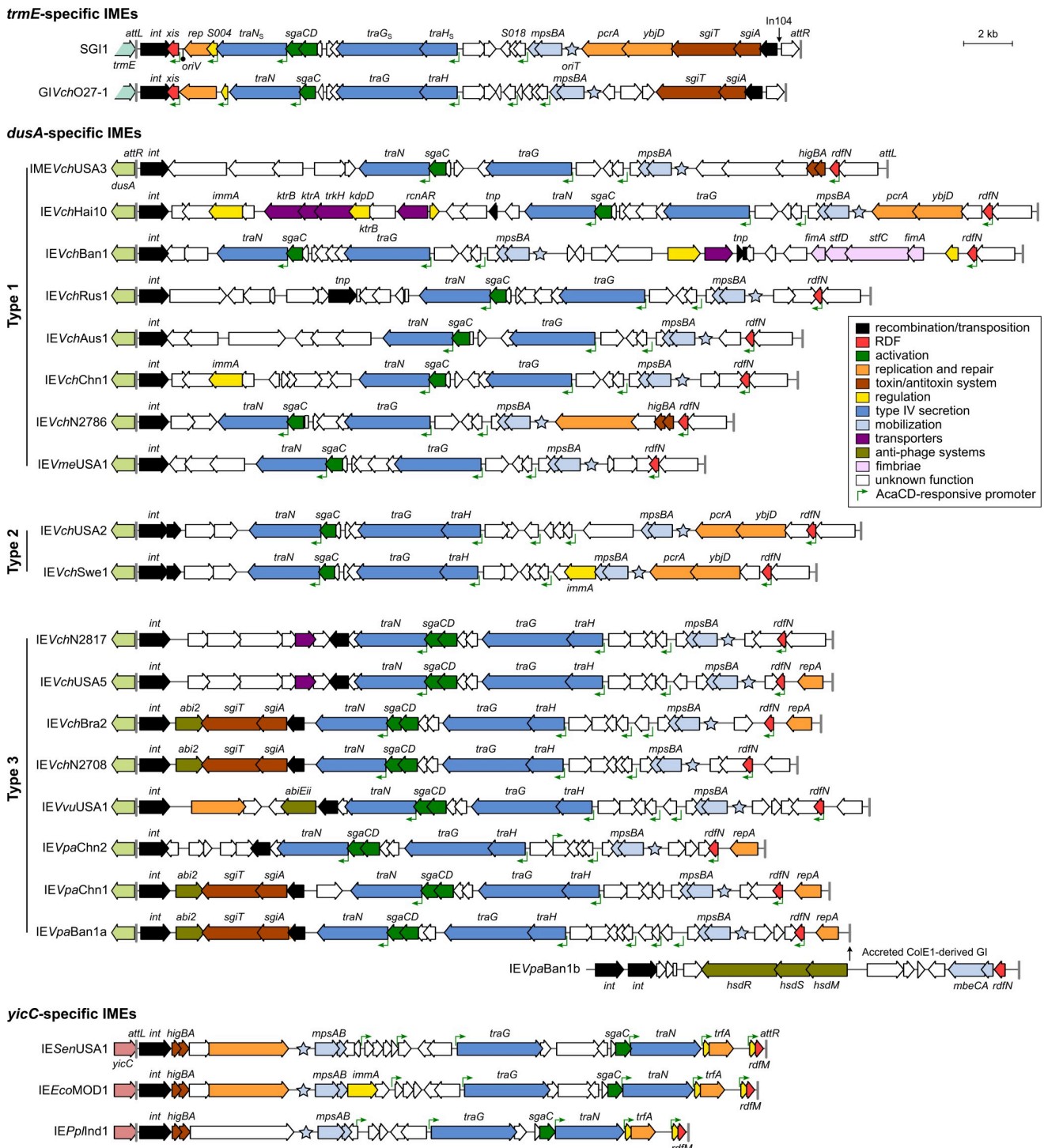

**Fig 1. Schematic representations of SGI1-related IEs.** The position and orientation of open reading frames (ORFs) are indicated by arrowed boxes. Colors depict the function deduced from functional analyses and BLAST comparisons. Potential AcaCD binding sites are represented by green angled arrows. Each island is flanked by the *attL* and *attR* (vertical grey lines) attachment sites when integrated into the 3' end of *trmE* (light blue), the 5' end of *dusA* (light green), or the 3' end of *yicC* (pink). The annotation of *attL* and *attR* relative to *int* is based on SGI1 (*trmE*) [9], IE*Aba*D1279779 of *Acinetobacter baumannii* D1279779 (*dusA*) [30] and MGI*Vfl*Ind1 (*yicC*) [3]. Details regarding ORFs are shown in S1 Dataset.

[7,9,12]. Thus far, the function of a few conserved genes has been characterized. Together with the *cis*-acting recombination site *attP*, the genes *int* and *xis* form the recombination module of SGI1 [13]. *int* encodes the site-specific tyrosine recombinase (integrase) that targets the 3' end of *trmE*. *xis* encodes the recombination directionality factor (RDF or excisionase) that enhances the excision reaction catalyzed by Int. The mobilization module includes the mobilization genes *mpsAB* and the *oriT* located upstream of *mpsA* [19]. *mpsA* encodes an atypical relaxase distantly related to tyrosine recombinases. Unlike most characterized IMEs, SGI1 carries a replicon composed of an iteron-based origin of replication (*oriV*) and the replication initiator gene *rep* [20,21]. SgaCD, a transcriptional activator complex expressed by SGI1 in response to a coresident IncC plasmid, controls *rep* expression [21,22]. The excised replicative form of SGI1 destabilizes the helper plasmid by an unknown process, and is further stabilized by its *sgiAT* addiction module [20,22–24]. Finally, SGI1 encodes three mating pore subunits, TraN$_S$, TraH$_S$, and TraG$_S$, that actively replace their counterparts in the T4SS encoded by the IncC plasmid [25]. The substitution of TraG allows SGI1 to bypass the IncC-encoded entry exclusion mechanism and transfer between cells carrying conjugative plasmids belonging to the same entry exclusion group [26].

Given the high similarity between SGI1 variants integrated at *trmE*, we undertook a search for distant SGI1-like IMEs in bacterial genomes using MpsA, TraG$_S$, SgaC, and TraN$_S$ as baits. Here, we report the existence of distantly related IMEs integrated at the 5' end of *dusA* in several species of *Vibrionaceae* and the 3' end of *yicC* in several species of *Gammaproteobacteria*. We have examined the interactions between an IncC plasmid and IME*Vch*USA3, a *dusA*-specific representative IME from an environmental *V. cholerae* strain. The genetic determinants required for the excision of IME*Vch*USA3 and its mobilization by IncC plasmids were characterized. Finally, we took a fresh look at the emergence and evolution of SGI1 and its siblings by conducting phylogenetic analyses and proposed a hypothetical evolutionary pathway of putative IMEs resembling SGI1.

## Results

### Novel integrative elements (IEs) distantly related to SGI1 are inserted in *dusA* and *yicC* in various *Gammaproteobacteria*

To find novel SGI1-like elements, we searched the Refseq database using blastp and the primary sequences of MpsA, TraG$_S$, SgaC, and TraN$_S$. Considering the substitution of integration modules can change the integration site [27–29], the integrase Int$_{trmE}$ was excluded from the analysis. We identified 24 distinct integrative elements encoding homologs of the four bait proteins in 36 different bacterial strains (Fig 1, Tables 1 and S1). 21 of these IEs are integrated into the 5' end of *dusA* (tRNA-dihydrouridine synthase A) in diverse *Vibrio* species from various origins. The remaining three are located at the 3' end of *yicC* (unknown function) in *E. coli*, *Aeromonas veronii*, *P. mirabilis*, *S. enterica* serovar Kentucky, and *Pokkaliibacter plantistimulans*. The size of the IEs varies from 22.8 kb to 37.1 kb. The conserved genes *mpsA* (together with *mpsB*), *traG*, *traN*, and *sgaC* remain in a syntenic order, though sporadically separated by variable DNA (Fig 1).

Consistent with the change of integration site, the respective *int* genes of SGI1 and the *dusA*- and *yicC*-specific IEs do not share any sequence similarity. Furthermore, unlike SGI1, these novel IEs lack *xis* downstream of *int* (Fig 1). Instead, *yicC*-specific IEs carry two small open reading frames (ORF) upstream of the *attR* site. The putative translation product of the second one shares 35% identity over 65% coverage with the excisionase RdfM of MGI*Vfl*Ind1 [31]. Although *dusA*-specific IEs lack *xis* and *rdfM*, all carry an ORF predicted to encode a 76-aminoacyl residue protein containing the pyocin activator protein PrtN domain (Pfam PF11112). Based on its size, position, predicted DNA-binding function, conservation, and evidence presented below, we named this ORF *rdfN*.

**Table 1. Main features of the IEs described in this study.**

| IE Name | Organism[1] | Size (bp) | Target site | Genbank accession number |
|---|---|---|---|---|
| GI*Vch*Rus1 | *V. cholerae* 1 | 30,204 | *dusA* | NZ_SMZE01000022 |
| IE*Vch*Aus1 | *V. cholerae* A12JL36W30 | 27,410 | *dusA* | NZ_VIOZ01000074 |
| IE*Vch*USA5 | *V. cholerae* OYP2C05 | 28,706 | *dusA* | NZ_NMTM01000021 |
| IME*Vch*USA3 | *V. cholerae* OYP6G08 | 30,910 | *dusA* | NZ_NMSY01000009 |
| IE*Vch*A215[2] | *V. cholerae* A215 sv Inaba | 29,933 | *dusA* | CWPR01000020.1 |
| IE*Vch*USA2 | *V. cholerae* 692–79 | 29,931 | *dusA* | MIPA01000024 |
| IE*Vch*N2751[2] | *V. cholerae* N2751 | 30,018 | *dusA* | NZ_VSGL01000012 |
| IE*Vch*N2744[2] | *V. cholerae* N2744 | 30,134 | *dusA* | NZ_VSGF01000021.1 |
| IE*Vch*N2708 | *V. cholerae* N2708 | 27,248 | *dusA* | NZ_VSFQ01000013 |
| IE*Vch*N2786 | *V. cholerae* N2786 | 24,658 | *dusA* | NZ_VSHP01000008 |
| IE*Vch*N2817 | *V. cholerae* N2817 | 28,717 | *dusA* | NZ_VSIM01000004 |
| IE*Vch*Chn1 | *V. cholerae* N2787 | 27,195 | *dusA* | NZ_VSHQ01000015 |
| IE*Vch*Ban1 | *V. cholerae* EM-1676-A | 36,519 | *dusA* | NZ_KB662834 |
| IE*Vch*Hai10 | *V. cholerae* 2012Env-2 | 37,162 | *dusA* | NZ_JSTD01000059/60 |
| IE*Vch*Swe1 | *V. cholerae* 11116 | 28,100 | *dusA* | NZ_MDYK01000006 |
| IE*Vch*Bra2 | *V. cholerae* TMA-21 | 28,230 | *dusA* | ACHY01000008 |
| IE*Vme*USA1 | *V. metoecus* 07–2435 | 23,518 | *dusA* | NZ_LCUE01000016 |
| IE*Vpa*Chn1 | *V. parahaemolyticus* GIMxtf41-2013.07 | 28,589 | *dusA* | NZ_MRWJ01000014 |
| IE*Vpa*Chn2 | *V. parahaemolyticus* C2_8 | 25,944 | *dusA* | NZ_NNLT01000047 |
| IE*Vpa*Ban1a | *V. parahaemolyticus* NIHCB0757 | 29,418 | *dusA* | AVPX01000004 |
| IE*Vvu*USA1 | *V. vulnificus* VA-WGS-18041 | 30,228 | *dusA* | NZ_RBZL01000019 |
| IE*Eco*MOD1 | *E. coli* MOD1-EC5437 | 25,611 | *yicC* | NZ_NLPO01000006 |
| IE*Sen*USA1 | *S. enterica* Kentucky ARS-CC8289 | 25,970 | *yicC* | NZ_MCPS01000044 |
| IE*Ppl*Ind1 | *P. plantistimulans* L1E11 | 22,777 | *yicC* | NZ_LAPT01000132/095 |
| GI*Vch*O27-1 | *V. cholerae* 10432–62 | 26,646 | *trmE* | CP010812 |

[1] *V.*, *Vibrio*; *E.*, *Escherichia*; *S.*, *Salmonella*; *P.*, *Pokkaliibacter*

[2] Not represented in Fig 1 as nucleotide sequences contain gaps

Phylogenetic analysis of Int$_{yicC}$ proteins of *yicC*-specific SGI1-like IEs form a cluster distinct from the integrases of IMEs mobilizable by IncC plasmids through a MobI protein (Pfam PF19456), such as MGI*Vmi*1, and IMEs that mimic the *oriT* of SXT/R391 ICEs, such as MGI*Vfl*Ind1 [3,17,32] (Fig 2A).

Phylogenetic analysis of Int$_{dusA}$ proteins confirmed that the integrases of these IEs form a monophyletic group exclusive to the *Vibrionaceae* and distinct from those encoded by other *dusA*-specific IEs found in other taxa, including GI*Aca*Bra1 from *Aeromonas caviae* that is likely mobilizable by IncC plasmids via a MobI protein [32] (Fig 2B). Int$_{dusA}$ proteins of the IEs identified here share at least 75% identity, while identities drop below 60% with the non-*Vibrio* Int$_{dusA}$ proteins (Fig 2C). Sequence logos built using alignments of the *attL* and *attR* chromosomal junctions revealed a 21-bp imperfect repeat at the extremities of each IE (Fig 2B). This repeat is similar to the one reported for *dusA*-specific IEs found in a broader range of species [30].

## Three types of *dusA*-integrated SGI1-related elements

Blastn and blastp analyses using SGI1ΔIn104 as the reference confirmed that the identified *dusA*-specific IEs share limited sequence similarities with SGI1 (S1A Fig). Besides the genes encoding MpsA, TraG, SgaC, and TraN, all carry the auxiliary mobilization factor gene *mpsB*

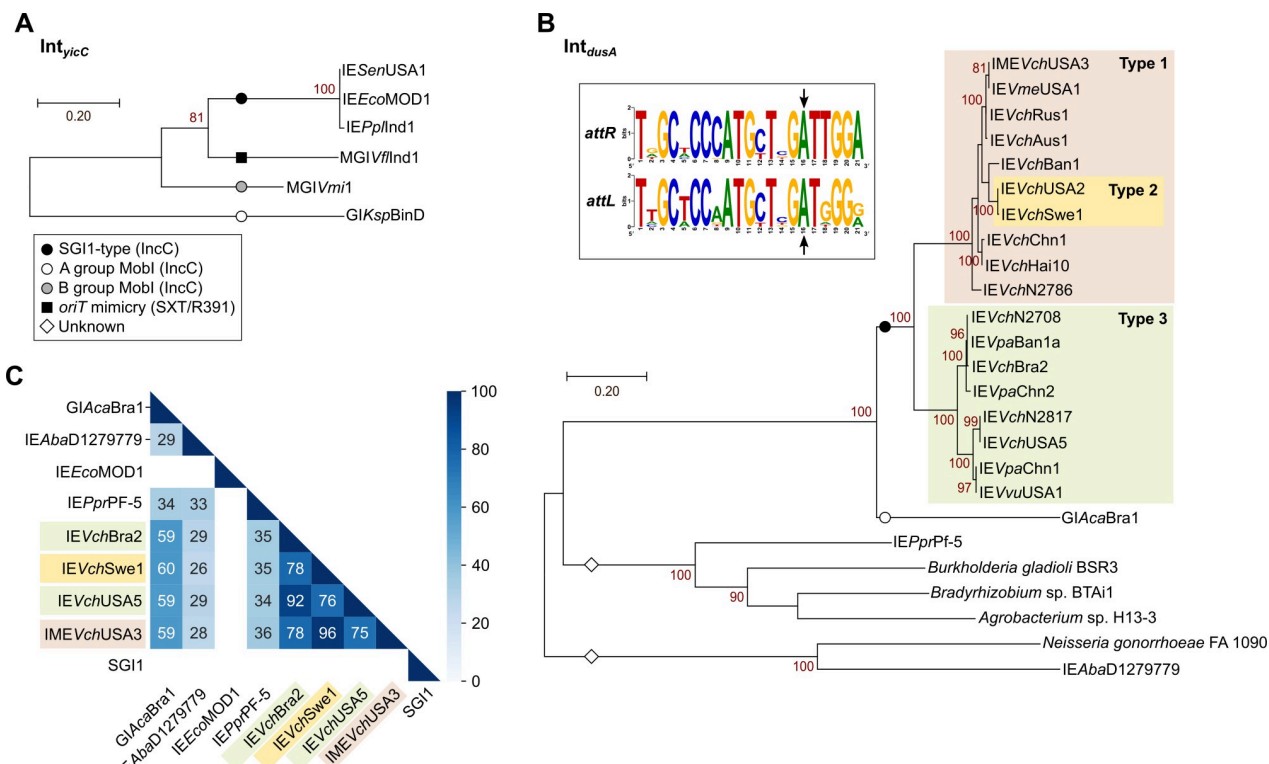

**Fig 2. Integrases encoded by the *yicC*- and *dusA*-specific IEs.** Maximum likelihood phylogenetic analyses of Int$_{yicC}$ (A) and Int$_{dusA}$ (B). The trees are drawn to scale, with branch lengths measured in the number of substitutions per site over 400 and 359 amino acid positions for Int$_{yicC}$, and Int$_{dusA}$, respectively. The helper elements and mechanism of mobilization are indicated for each lineage according to the keys shown in the legend box of panel A. The inset of panel B shows logo sequences of the repeats in *attL* and *attR* attachment sites. The arrows indicate the island termini experimentally determined for IE*Aba*D1279779 by Farrugia *et al.* [30]. (C) Heatmap showing blastp identity percentages of pairwise comparison of Int$_{dusA}$ representative proteins. Proteins accession numbers are provided in S2 Dataset, except for IE*Aba*D1279779 (WP_000534871.1), IE*Ppr*Pf-5 of *Pseudomonas protegens* Pf-5 (WP_011060295.1), and IEs of *Burkholderia gladioli* BSR3 (WP_013697845.1), *Bradyrhizobium* sp. BTAi1 (WP_012043559.1), *Agrobacterium* sp. H13-3 (WP_013636109.1), and *Neisseria gonorrhoeae* FA 1090 (EFF39980.1).

and the *oriT* sequence (Fig 1). Secondary structure prediction of the aligned *oriT* sequences located upstream of *mpsA* using RNAalifold revealed that despite the sequence divergence, the structure of *oriT* with three stem-loops was strictly conserved (S2B Fig). In contrast, *sgaD* is not strictly conserved and highly divergent from *sgaD* of SGI1 when present (Figs 1 and S1A).

Comparison using IE*Vch*USA2 as the reference suggests that *dusA*-specific IEs cluster into three distinct types as confirmed by the phylogenetic analysis of concatenated MpsA-TraG-SgaC-TraN (Figs 3A, S1B, and S3). Type 1 *dusA*-specific IEs such as IME*Vch*USA3 are mainly found in *V. cholerae* and lack both *traH* and *sgaD* (Figs 1 and 3A). Type 2 IEs such as IE*Vch*USA2 lack *sgaD* but carry *traH*. This lineage only includes two *dusA*-specific IEs of *V. cholerae* but also closely related *yicC*-specific IEs such as IE*Eco*MOD1 and the *trmE*-specific GI*Vch*O27-1. Finally, type 3 IEs such as GI*Vch*USA5 are the most distant from the two other types and SGI1. Type 3 IEs carry both *traH* and *sgaD* and reside in diverse *Vibrio* species. With the exception of a few outliers encoded by IEs such as IE*Vch*N2817, IE*Vch*N2708 or IE*Ppl*Ind1, the proteins MpsA, TraG, SgaC and TraN encoded by members of the same type typically share more than 95% identity (Figs 3B and S3). MpsA remains the least divergent protein between the three types, sharing at least 65% identity between type 1 and type 3, and from 64% to 93% with SGI1. In contrast, TraG and TraN are the most divergent between types, ranging from 46% to 59% for TraG and from 46% to 76% for TraN.

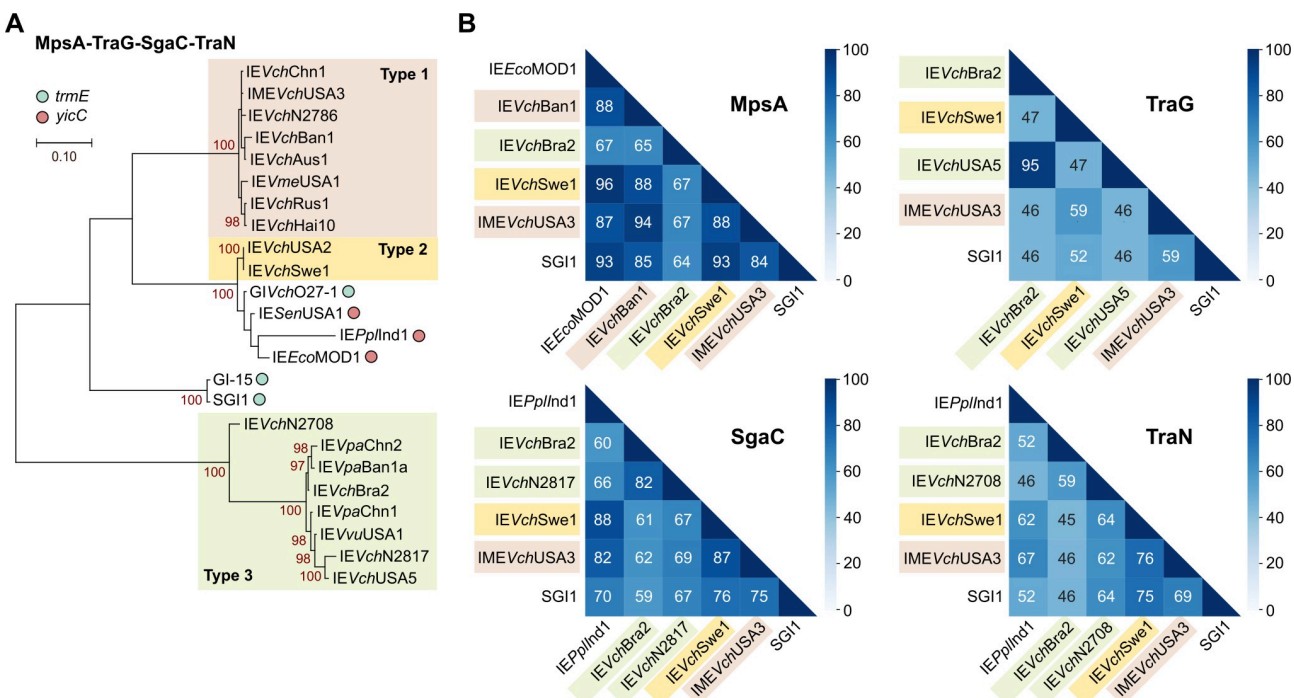

**Fig 3. Conserved genes support three main lineages of *dusA*-specific SGI1-like IEs.** (A) Maximum likelihood phylogenetic analysis of concatenated MpsA-TraG-SgaC-TraN. The tree is drawn to scale, with branch lengths measured in the number of substitutions per site over 2,637 amino acid positions. Taxa corresponding to IEs targeting *trmE* and *yicC* are indicated by a light blue circle and a red circle, respectively. All other taxa correspond to *dusA*-specific IEs. Phylogenetic relationships of MpsA, TraG, SgaC and TraN proteins are shown separately in S2 Fig. (B) Heatmaps showing blastp identity percentages of pairwise protein comparisons for representatives of MpsA, TraG, SgaC, and TraN. Proteins accession numbers and clusters are provided in S1 Table and S2 Dataset.

Worthy of note, these three distinct lineages of *dusA*-specific IEs are supported by the phylogeny of the *oriT* sequences (S2A Fig). Again, *oriT* loci of type 3 IEs strongly diverge from those of types 1 and 2, as well as from the *oriT* loci of the highly homogenous SGI1 group.

## Variable features found in the *dusA*- and *yicC*-specific IEs

Most variable genes in the identified IEs encode proteins of unknown function. A search for antibiotic resistance determinants using the Resistance Gene Identifier server failed to reveal any known resistance gene. Several IEs encode putative functions altering host processes and virulence, including the transport of ions and small molecules (*ktrAB*, *trkH*, and *kdpD* for potassium uptake and *rcnAR* for nickel/cobalt efflux in IE*Vch*Hai10, sulfite export in IE*Vch*N2817 and IE*Vch*Swe1), c-di-GMP degradation (IE*Vch*Ban1), and fimbriae (IE*Vch*-Ban1) (S1 Dataset).

None of the reported IEs carries the same replication module (*S004-rep-oriV*) as canonical SGI1. Instead, five *dusA*-specific IEs belonging to the type 3 lineage (IE*Vch*USA5, IE*Vch*Bra2, IE*Vpa*Chn1, IE*Vpa*Chn2, and IE*Vpa*Ban1a) encode a putative replication initiator protein with the IncFII_repA domain (Pfam PF02387) (Fig 1, S1 Dataset). IE*Vvu*USA1 encodes a putative helicase with an UvrD_C_2 domain (Pfam PF13538), whereas, like SGI1, IE*Vch*-Hai10, IE*Vch*USA2 and IE*Vch*Swe1 encode a putative ATP-dependent helicase (PcrA) and a putative ATP-dependent endonuclease (YbjD). In addition, IE*Vch*N2786, IE*Sen*USA1 and IE*Eco*MOD1 encode a predicted DEAD/DEAH box helicase (Pfam PF00270 and PF00271). The three *yicC*-specific IEs encode a homolog of TrfA (Pfam PF07042), the replication initiator

protein of broad-host-range IncP plasmids [33]. No replicative functions could be ascribed with confidence to any gene carried by the other *dusA*-specific IEs. Several IEs also encode toxin-antitoxin systems, such as *sgiAT* and *higAB*, which likely enhance their stability (Fig 1). In the type 3 IEs IE*Vch*Bra2, IE*Vch*N2708, IE*Vpa*Chn1 and IE*Vpa*Ban1a, *sgiAT* is associated with a gene coding for a putative abortive infection bacteriophage resistance factor (Abi_2, Pfam PF07751). Likewise, IE*Vvu*USA1 carries a gene coding for a different putative abortive infection bacteriophage resistance factor (AbiEii toxin, Pfam PF13304).

Finally, IE*Vpa*Ban1a is integrated at *dusA* adjacent to a distinct IE, IE*Vpa*Ban1b, in a tandem fashion. GI*Vpa*Ban1b codes for two predicted integrases sharing 44% and 27% identity with $Int_{dusA}$ of IE*Vpa*Ban1a. GI*Vpa*Ban1b encodes a putative type I restriction-modification system, a MobA-like relaxase ($MOB_{P1}$), the mobilization auxiliary factor MobC, and an RdfN homolog (Fig 1).

## Non-canonical SGI1-like IEs carry AcaCD-responsive genes

Considering the divergence of the 24 new IEs from prototypical SGI1, we wondered whether an IncC plasmid could mobilize them like SGI1. The hallmark of IncC-dependent mobilization is the presence of AcaCD-responsive promoters in IncC-mobilizable IEs. Hence, we searched for putative AcaCD-binding sites in the sequences of *trmE*-specific IEs (prototypical SGI1 was used as the positive control) and the *yicC*- and *dusA*-specific IEs. In these IEs, an AcaCD-binding motif was predicted upstream of *traN*, *traHG* (or *traG*), *S018*, and *xis* (or *rdfM* or *rdfN*) (Figs 1 and S4). Moreover, an AcaCD-binding motif was also predicted upstream of *trfA* in the *yicC*-specific IEs.

We cloned the promoter sequences of *int*, *traN*, *traG*, *S018*, and *rdfN* of IME*Vch*USA3 upstream of a promoterless *lacZ* reporter gene and monitored the β-galactosidase activity with or without AcaCD. The promoter $P_{int}$ was active regardless of the presence of AcaCD (Fig 4A). In contrast, the four other promoters exhibited weak activity in the absence of AcaCD. Upon induction of *acaDC* expression, $P_{traN}$ and $P_{S018}$ remained unresponsive, while the activities of $P_{traG}$ and $P_{rdfN}$ increased 40 and 400 times, respectively (Fig 4B). The inertia of $P_{traN}$ and $P_{S018}$ toward AcaCD could result from single nucleotide substitutions in the AcaCD binding site previously shown to be essential for recruiting the activator [22]: CCS**A**AAWW instead of CCS**C**AAWW in $P_{traN}$ and CCC**C**AAAA instead of CCC**A**AAAA in $P_{S018}$ (S4 Fig).

Hence, despite their divergence and different integration sites, these IEs share with SGI1 a common activation mechanism elicited by the presence of an IncC plasmid.

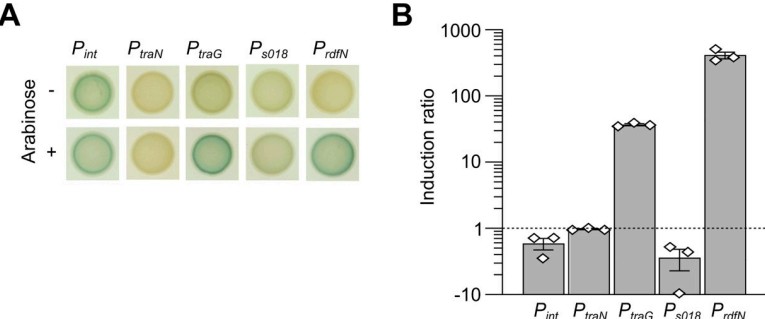

**Fig 4. β-galactosidase activities of the promoters $P_{int}$, $P_{traN}$, $P_{traG}$, $P_{S018}$ and $P_{rdfN}$ of IME*Vch*USA3 transcriptionally fused to *lacZ*.** (A) Colonies were grown on LB agar with or without arabinose to induce *acaDC* expression from pBAD-*acaDC*. (B) Induction levels of the same promoters in response to AcaCD. β-galactosidase assays were carried out using the strains of panel A. Ratios between the enzymatic activities in Miller units for the arabinose-induced versus non-induced strains containing pBAD-*acaDC* are shown. The bars represent the mean and standard error of the mean of three independent experiments.

## IncC plasmids induce the excision and mobilization of IME*Vch*USA3

Next, we tested whether a coresident IncC plasmid could trigger the excision of IME*Vch*USA3 from *dusA* in its original host, *V. cholerae* OY6PG08. The derepressed IncC plasmid pVCR94<sup>Kn</sup> Δ*acr2* [34] was introduced into OY6PG08 by conjugation from *E. coli* KH40. The Δ*acr2* mutation improves the efficiency of interspecific transfer of the plasmid [35]. OY6PG08 Kn<sup>R</sup> transconjugants were tested by PCR to amplify the *attL* and *attR* chromosomal junctions, as well as the *attB* and *attP* sites resulting from the excision of IME*Vch*USA3 (S5A Fig). IME*Vch*USA3 was rarely retained in the transconjugants compared to the control IncC-free OY6PG08 clones, suggesting it was unstable and rapidly lost in IncC⁺ cells (S5B and S5C Fig).

To test the interspecific mobilization of IME*Vch*USA3 from *V. cholerae* OY6PG08, we inserted a selection marker upstream of *traG* and used pVCR94<sup>Kn</sup> Δ*acr2* as the helper plasmid. IME*Vch*USA3<sup>Cm</sup> transferred to *E. coli* CAG18439 at a frequency of $7.01 \times 10^{-5}$ transconjugant/donor CFUs. Amplification of *attL* and *attR* using *E. coli*-specific primers confirmed that IME*Vch*USA3 integrates at *dusA* in *E. coli* (S5D Fig).

## Excision of *dusA*-specific IEs depends on *rdfN*

To further characterize the biology of IME*Vch*USA3, we measured its excision rate and copy number by qPCR, with and without coresident pVCR94<sup>Sp</sup>. We also monitored its intraspecific transfer (*E. coli* to *E. coli*) in the same context. Spontaneous excision of the IE rarely occurred ($<0.001\%$ of the cells) (Fig 5A). In contrast, in the presence of the helper plasmid, the free *attB* site was detected in more than 67% of the cells confirming that the IncC plasmid elicits the excision of IME*Vch*USA3<sup>Kn</sup>. Likewise, the presence of the plasmid resulted in a ~3-fold increase of the copy number of IME*Vch*USA3<sup>Kn</sup> (Fig 5B), suggesting that the excised form undergoes replication. The frequency of transfer of IME*Vch*USA3<sup>Kn</sup> was comparable to that of the helper plasmid (~3.5×10⁻² transconjugants/donor), while the frequency of cotransfer was more than two logs lower (Fig 5C).

Thus far, the factors required to catalyze the excision of *dusA*-specific IEs have not been examined [30]. Whereas all *dusA*-specific IEs lack *xis* downstream of *int*, they carry a small ORF, here named *rdfN*, coding for a putative PrtN homolog (Fig 1) [30]. The deletion of *rdfN* abolished the excision and replication of IME*Vch*USA3<sup>Kn</sup>. Complementation by ectopic expression of *rdfN* from the arabinose-inducible promoter $P_{BAD}$ restored the wild-type excision level but not the replication (Fig 5A and 5B). Likewise, deletion of *rdfN* abolished the mobilization of IME*Vch*USA3<sup>Kn</sup> but had no impact on the transfer of the helper plasmid (Fig 5C), confirming the specific role of *rdfN* in the IE's mobility.

To confirm that *rdfN* encodes the sole and only RDF of IME*Vch*USA3, we constructed mini-IE, a minimal version of IME*Vch*USA3 that only contains *int* and a spectinomycin-resistance marker. mini-IE is flanked by *attL* and *attR* and is integrated at *dusA* in *E. coli* EC100 (Fig 5D). Using mini-IE, *attB* and *attP* were detected only upon ectopic expression of *rdfN* from pBAD-*rdfN*, confirming that no other IME*Vch*USA3-encoded protein besides Int and RdfN is required for the excision of the element (Fig 5E). *rdfN* is the essential RDF gene that favors the excision of IME*Vch*USA3 and, most likely, all *dusA*-specific IEs.

## A SgaC/AcaD hybrid complex activates the excision and mobilization of IME*Vch*USA3

Next, we investigated the role of the transcriptional activator genes *acaC* and *sgaC* in the mobilization of IME*Vch*USA3. Deletion of *acaDC* abolished the excision and replication of IME*Vch*USA3<sup>Kn</sup>, confirming that its excision relies on *rdfN*, whose expression is activated by AcaCD

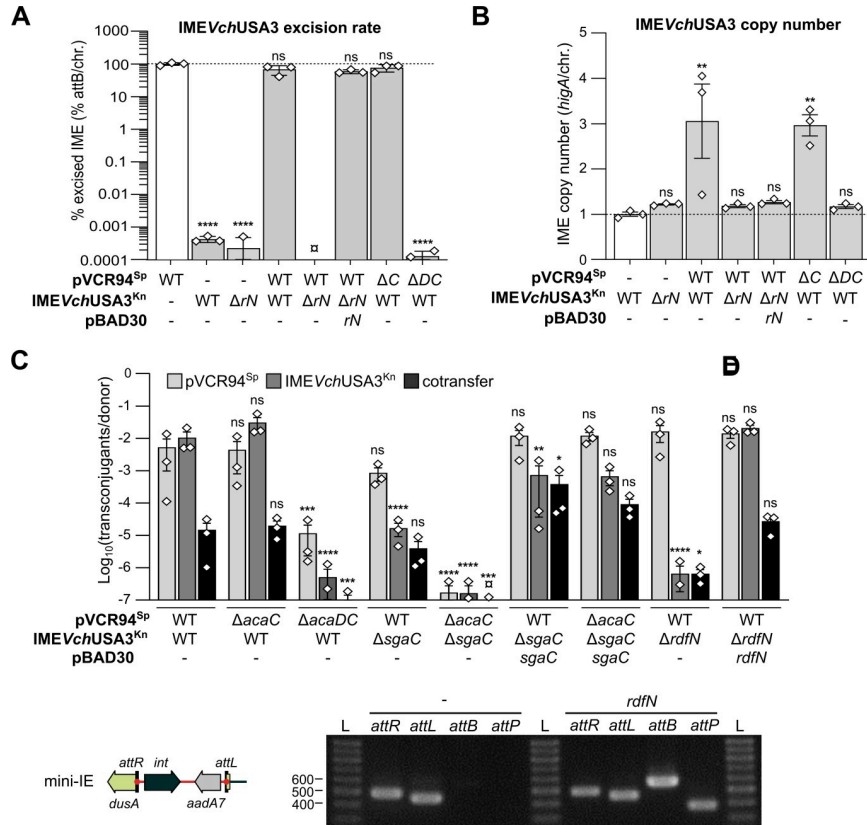

**Fig 5. Effect of *acaDC* and *rdfN* on the IncC-dependent excision and mobilization of IME*Vch*USA3.** (A) IME*Vch*USA3^Kn excision rate corresponds to the *attB*/chromosome ratio. (B) IME*Vch*USA3^Kn copy number corresponds to the *higA*/chromosome ratio. For panels A and B, all ratios were normalized using the control set to 1 and displayed in white. (C) Impact of *acaC*, *acaDC*, *sgaC* and *rdfN* deletions on the mobilization of IME*Vch*USA3. Conjugation assays were performed with CAG18439 (Tc) containing the specified elements as donor strains and VB112 (Rf) as the recipient strain. The bars represent the mean and standard error of the mean obtained from a biological triplicate. ¤ indicates that the excision rate or transfer frequency was below the detection limit. Statistical analyses were performed (on the logarithm of the values for panels A and C) using a one-way ANOVA with Dunnett's multiple comparison test. For panels A and B, statistical significance indicates comparisons to the normalization control. Statistical significance is indicated as follows: ****, $P < 0.0001$; ***, $P < 0.001$; **, $P < 0.01$; *, $P < 0.05$; ns, not significant. (D) Schematic representation of mini-IE inserted at the 5' end of *dusA*. (E) RdfN acts as a recombination directionality factor. Detection of *attB*, *attP*, *attL* and *attR* sites by PCR in colonies of *E. coli* EC100 *dusA*::mini-IE in the presence or absence of *rdfN*. L, 1Kb Plus DNA ladder (Transgen Biotech).

(Figs 4, 5A, and 5B). The mutation also confirmed that SgaC provided by IME*Vch*USA3^Kn is insufficient by itself to elicit *rdfN* expression. The excision rate remained extremely low in cells that lack the helper plasmid or cells that carry pVCR94^Sp Δ*acaDC*. However, IME*Vch*USA3^Kn allowed the low-frequency transfer of pVCR94^Sp Δ*acaDC* [17] (Fig 5C). Hence SgaC alone can activate to some degree the expression of the transfer genes of the helper plasmid. In contrast, deletion of *acaC* had no significant impact on the excision, replication, and mobilization of IME*Vch*USA3^Kn, or on the transfer of the helper plasmid (Fig 5A, 5B, and 5C). The primary sequences of AcaC and SgaC from IME*Vch*USA3 share 85% identity over 94% coverage, whereas AcaC and SgaC from SGI1 share only 75% identity over 92% coverage. Hence AcaD produced by the plasmid and SgaC produced by the IME likely generate a functional chimeric transcriptional complex that acts as a potent activator of *rdfN* and the transfer genes.

The transfer of IME*Vch*USA3^Kn Δ*sgaC* decreased nearly 3 logs compared to the wild-type IE, despite the presence of *acaDC* on the helper plasmid (Fig 5C). Moreover, deletion of both

*acaC* and *sgaC* nearly abolished all transfer. Ectopic expression of *sgaC* alone from pBAD-*sgaC* complemented these deletions to wild-type levels (Fig 5C). These observations confirm that *sgaC*, not *acaC*, combined with *acaD* produces a hybrid activator complex that is essential for the excision and mobilization of IME*Vch*USA3.

## IME*Vch*USA3 provides a new promoter and N-terminus for *dusA* expression

Since *dusA*-specific IEs insert within the 5' end of *dusA*, we wondered whether the gene remains expressed after the integration event. Sequence analysis of the *attR* junction of *E. coli* K12 transconjugants revealed that IME*Vch*USA3 provides a new 5' coding sequence that diverges significantly from the native *E. coli dusA* gene (Fig 6A). This alteration of the 5' end of *dusA* results in a novel N-terminus of identical length sharing 61% identity over the 35 initial amino acid residues with native DusA. To test the expression of *dusA*, we constructed a translational *lacZ* fusion to its fortieth codon downstream of the *attR* junction in *E. coli* CAG18439 and BW25113 (Fig 6B). β-galactosidase assays revealed that *dusA* remains expressed after integration in both strains, confirming that IME*Vch*USA3 provides a new promoter (Fig 6C). However, we observed a statistically significant reduction of *dusA* expression resulting from the integration of the IE in both strains, suggesting that the transcription or translation signals brought by the IE are weaker than the original ones upstream of *E. coli dusA*.

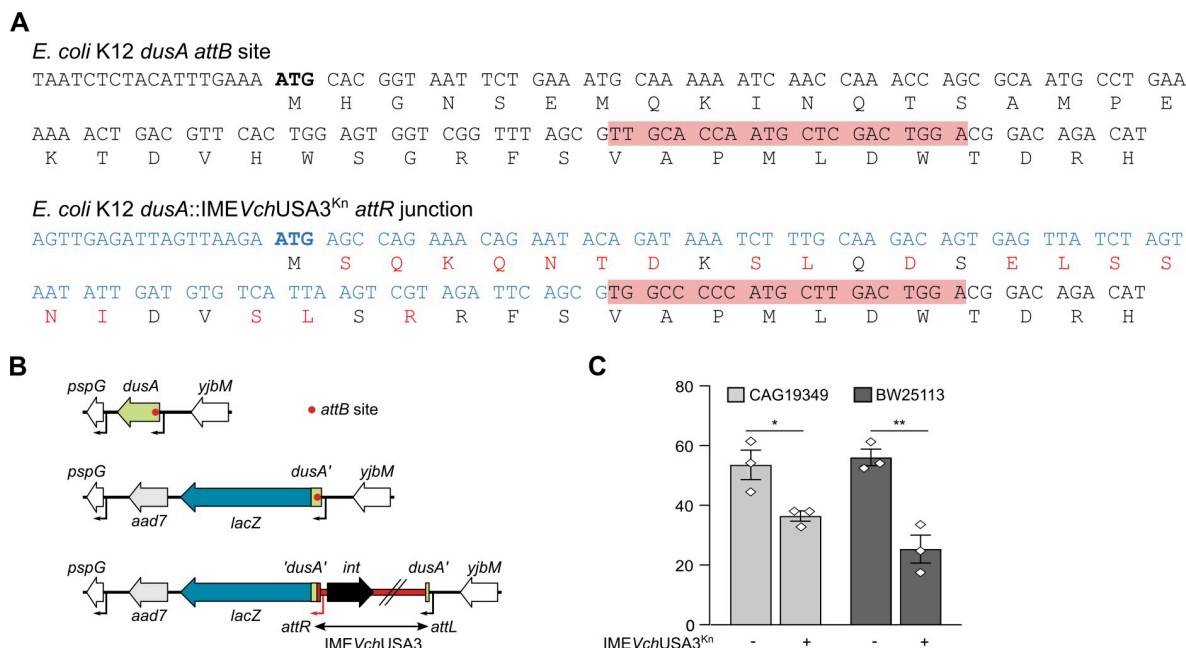

**Fig 6. IME*Vch*USA3 drives the expression of *dusA*.** (A) Comparison of the coding sequences of the 5' end of *dusA* in *E. coli* K12 MG1655 before (*attB* site) and after (*attR* junction) the integration of IME*Vch*USA3. The core sequence of the *attB* and *attR* recombination sites is indicated with red shading. The ATG start codon of *dusA* is shown in bold. The sequence shown in blue is internal to IME*Vch*USA3. Amino acid residues shown in red differ from the native N-terminus of DusA. This sequence was obtained by sequencing the *attR* junction of an *E. coli* CAG18439 *dusA*::IME*Vch*USA3^Kn transconjugant colony. (B) Schematic representation of the *dusA'-lacZ* translational fusion for the detection of *dusA* expression. (C) β-galactosidase activity of the *dusA'-'lacZ* fusion before (-) and after (+) insertion of IME*Vch*USA3^Kn in *E. coli* CAG18439 (FD034) and BW25113 (FD036). The bars represent the mean and standard error of the mean of three independent experiments. Statistical analyses were performed using an unpaired t test to compare the expression before and after integration of IME*Vch*USA3^Kn for each strain. Statistical significance is indicated as follows: **, $P < 0.01$; *, $P < 0.05$.

## Discussion

SGI1-like elements integrated at the 3' end of *trmE* are widespread in a broad range of *Entero-bacteriaceae* and sporadically found in a few *Vibrio* species [7]. The integrase of SGI1 and its variants occasionally targets the intergenic region between *sodB* and *purR* genes, a secondary attachment site [36]. Here, we report the identification of distant SGI1-like elements that specifically target the 5' end of *dusA* in multiple *Vibrio* species and the 3' end of *yicC* in *Enterobacteriaceae* and *Balneatrichaceae*. Farrugia *et al.* [30] already described IEs integrated at the 5' end of *dusA*, mostly prophages or phage remnants found exclusively in *Alpha-*, *Beta-* and *Gammaproteobacteria*. These authors identified IE*Vch*Ban1 and IE*Vch*Bra2 in *V. cholerae*, and several other IEs predicted to encode conjugative functions in *Bradyrhizobium*, *Caulobacter*, *Mesorhizobium*, *Paracoccus*, *Pseudomonas*, and *Rhodomicrobium* [30]. Our group recently reported a *dusA*-specific IE in *Aeromonas caviae* 8LM potentially mobilizable by IncC plasmids [32]. GIA-ca8LM lacks *tra* genes but encodes a mobilization protein (MobI) under the control of an AcaCD-responsive promoter. Together, these reports confirm that *dusA* is an insertion hotspot for distinct families of mobile elements across at least three *Proteobacteria* phyla.

Thus far, only the *dusA*-specific IEs in *A. baumannii* D1279779 and *P. protegens* Pf-5 were shown to excise from the chromosome, albeit at a low level [30]. Neither IE has been tested for intercellular mobility. Here, we characterized IME*Vch*USA3, a representative member of a subgroup of *dusA*-specific IEs circulating in *Vibrio* species and distantly related to SGI1. We demonstrated that IME*Vch*USA3 is mobilizable by IncC conjugative plasmids to *E. coli*. In the presence of an IncC plasmid, this IME excises in practically all cells of the population and becomes highly unstable (Figs 5A and S5B). We showed that its excision was under the control of AcaCD provided by the IncC plasmid and required *rdfN*, a gene whose expression is driven by an AcaCD-responsive promoter (Fig 4). *rdfN* encodes a novel RDF distantly related to the pyocin activator protein PrtN of *Pseudomonas*. *rdfN* seems to be ubiquitous, yet highly divergent, in *dusA*-specific IEs reported by Farrugia *et al.* [30]. For instance, RdfN (PrtN) encoded by the IE of *P. protegens* Pf-5 shares only 29% identity with RdfN of IME*Vch*USA3, and their promoters are unrelated. Hence, the expression of *rdfN* homologs encoded by different families of *dusA*-specific IEs is likely controlled by different factors. Only the IEs that have evolved AcaCD-responsive promoters for their RDF gene are expected to be mobilizable by IncC or related plasmids.

Excision and mobilization of IME*Vch*USA3 occurred in the presence of a Δ*acaC* but not a Δ*acaDC* mutant of the helper plasmid (Fig 5), confirming that *sgaC* of the IME produces a functional activator subunit that can interact with AcaD provided by the plasmid. Furthermore, we showed here that, unlike *acaC*, *sgaC* plays a central role in the biology of IME*Vch*USA3 as the absence of *acaC* had no effect on the excision or transfer of the IME, while the absence of *sgaC* in spite of the presence of *acaC*, compromised its mobilization (Fig 5A, 5B, and 5C). We recently showed that AcaD most likely stabilizes the binding of AcaC to the DNA [22]. Therefore, AcaD and SgaC from IME*Vch*USA3 likely interact to form a chimerical activator complex. This interaction could compensate for the loss of *sgaD* in *yicC-* and type 1 and 2 *dusA*-specific IEs (Fig 1). The primary sequences of AcaC and SgaC of IME*Vch*USA3 (type 1) share 85% identity. In contrast, AcaC only shares 75% identity with SgaC of SGI1 and 64% identity with SgaC of GI*Vch*USA5 (type 3), suggesting that retention of *sgaD* allowed faster divergence of SgaC from AcaC. Retention of *sgaC* in the IEs could result from its essential role as the elicitor of excision and replication reported for SGI1. Indeed, although AcaCD binds to the promoters $P_{xis}$ and $P_{rep}$ of SGI1, it fails to initiate transcription at these two promoters, unlike SgaCD [22]. Nonetheless, $P_{xis}$ and $P_{rep}$ are not conserved in the IEs described here. *S004-rep* is missing, whereas *rdfN* or *rdfM* replaced *xis*, under the control of novel AcaCD-

responsive promoters (Figs 4 and S4). This observation raises intriguing questions regarding the recruitment of functional gene cassettes and their assimilation in a regulatory pathway. How did *xis*, *rdfN*, and *rdfM* acquire their AcaCD-responsive promoters? Is it by convergent evolution? What are the signals driving *rdfN* expression and IE excision in *dusA*-specific IEs resembling prophages?

Approximately 3 copies per cell of IME*Vch*USA3 were detected in the presence of the helper IncC plasmid (Fig 5B), lower than the copy number reported for SGI1 (~8 copies/cell) [20,22]. IME*Vch*USA3 lacks SGI1's replication module (*S004-rep-oriV*); however, one of the multiple genes of unknown function could encode an unidentified replication initiator protein. Notably, GI*Vch*O27-1 encodes a putative replication protein with an N-terminal replicase domain (PF03090) and a C-terminal primase domain (PriCT-1, PF08708) [20]. Multiple IEs also carry putative replicons based on *repA* and *trfA* (Fig 1), suggesting that independent replication is crucial in their lifecycle, perhaps to improve their stability in the presence of a helper plasmid [20–24].

Farrugia *et al.* [30] hypothesized that *dusA*-specific IEs could restore the functioning of DusA. We demonstrated here that IME*Vch*USA3 provides a new promoter allowing expression of *dusA*, though at a lower level than in IME-free cells, and restores the open reading frame with an altered N-terminus (Fig 6). Similarly, the ICE SXT that targets the 5' end of the peptide chain release factor 3 (RF3) gene *prfC* provides a new promoter and N-terminus in both *V. cholerae*, its original host, and *E. coli* [37]. In both cases, the consequences of the alteration of the N-terminus on the activity of the protein remain unknown.

The relative positions of *int* and *rdfN*/*rdfM* across the *attP* site suggest that, to remain functional, the recombination modules must be acquired or exchanged when the IEs are in their excised circular form. The promiscuity of different families of IEs targeting *yicC*, *dusA*, and *trmE* and mobilizable by IncC plasmids could act as the catalyst for these recombination events. During entry into a new host cell by conjugation, IncC plasmids elicit the excision of such IEs and promote homologous recombination between short repeated sequences in response to double-stranded break induced by host defense systems (CRISPR-Cas3) [34]. Hence the diversification of IncC plasmid-mobilizable IEs could be a side effect of the DNA repair mechanism used by these plasmids.

Unlike SGI1 and its siblings, all *dusA*-specific SGI1-like IEs reported here lack antibiotic resistance genes. Furthermore, SGI1 variants are prevalent in several pathogenic species and relatively well-conserved, whereas their *dusA*-specific relatives are scarce and highly divergent. These observations suggest that despite the considerable functional resemblances between *trmE*- and *dusA*-specific SGI1-like IEs, the epidemiological success of the SGI1 lineage has directly stemmed from the acquisition of class I integrons conferring multidrug resistance by forerunner elements such as SGI0 [38]. Based on the phylogenetic relationships between the core proteins MpsA, TraG, SgaC and TraN, *oriT* loci, and integrase proteins (Figs 2, 3, S2A, and S3), we propose a hypothetical evolutionary pathway leading to the emergence of the different types of IEs described here (Fig 7). The diversity of *dusA*-specific IEs and relative homogeneity of the SGI1 group suggest that the latter originated from the progenitor of IncA and IncC plasmids via a *dusA*-specific IE intermediate.

## Materials and methods

### Bacterial strains and media

Bacterial strains and plasmids used in this study are described in Table 2. Strains were routinely grown in lysogeny broth at 37°C in an orbital shaker/incubator and were preserved at -75°C in LB broth containing 20% (vol/vol) glycerol. Antibiotics were used at the following concentrations: ampicillin (Ap), 100 μg/ml; chloramphenicol (Cm), 20 μg/ml; erythromycin

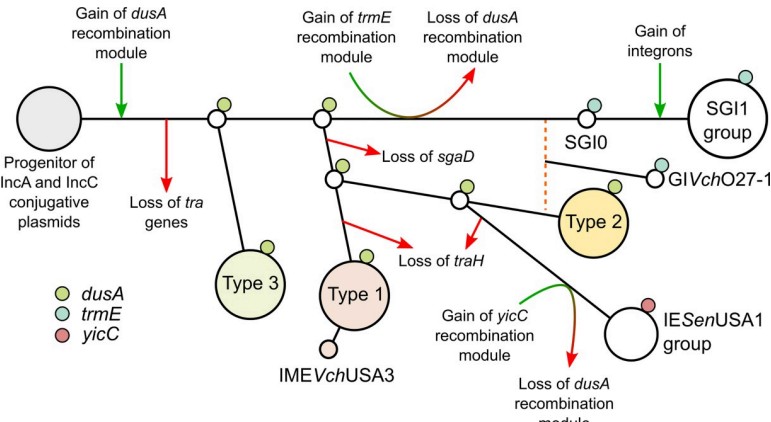

**Fig 7. Proposed hypothetical evolutionary pathway of SGI1-like IEs.** The sequence of events was inferred from the phylogenetic trees presented in this study, site of integration and conservation of *traH* and *sgaD* in the IEs. The proposed pathway ignores the gene cargo and presumes that the IE lineages evolved from the progenitor of IncA and IncC plasmids. The *dusA*-specific recombination module was chosen as the progenitor to minimize gain/loss and recombination events. Green and red arrows indicate gene gains and losses, respectively. The orange dashed line indicates a probable recombination event from which stemmed GI*Vch*O27-1.

(Em), 200 μg/ml; kanamycin (Kn), 10 μg/ml for single-copy integrants of pOP*lacZ*-derived constructs, 50 μg/ml otherwise; nalidixic acid (Nx), 40 μg/ml; rifampicin (Rf), 50 μg/ml; spectinomycin (Sp), 50 μg/ml; tetracycline (Tc), 12 μg/ml. Diaminopimelate (DAP) was supplemented to a final concentration of 0.3 mM when necessary.

## Mating assays

Conjugation assays were performed as previously described [25]. However, mixtures of donor and recipient cells were incubated on LB agar plates at 37˚C for 4 hours. Donors and recipients were selected according to their sole chromosomal markers. When required, mating experiments were performed using LB agar plates supplemented with 0.02% arabinose to induce expression of pBAD30-derived complementation vectors. Frequencies of transconjugant formation were calculated as ratios of transconjugant per donor CFUs from three independent mating experiments.

## Molecular biology

Plasmid DNA was prepared using the QIAprep Spin Miniprep Kit (Qiagen), according to manufacturer's instructions. Restriction enzymes used in this study were purchased from New England Biolabs. Q5 DNA polymerase (New England Biolabs) and EasyTaq DNA Polymerase (Civic Bioscience) were used for amplifying cloning inserts and verification, respectively. PCR products were purified using the QIAquick PCR Purification Kit (Qiagen), according to manufacturer's instructions. *E. coli* was transformed by electroporation as described by Dower *et al.* [39] in a Bio-Rad GenePulser Xcell apparatus set at 25 μF, 200 Ω and 1.8 kV using 1-mm gap electroporation cuvettes. Sanger sequencing reactions were performed by the Plateforme de Séquençage et de Génotypage du Centre de Recherche du CHUL (Québec, QC, Canada).

## Plasmids and strains constructions

Plasmids and oligonucleotides used in this study are listed in Tables 2 and S2, respectively. IME*Vch*USA3^Cm^ was constructed by inserting the *pir*-dependent replication RP4-mobilizable

**Table 2. Strains and elements used in this study.**

| Strains or elements | Relevant genotype or phenotype[a] | Source or reference |
|---|---|---|
| *V. cholerae* | | |
| OY6PG08 | Environmental, Oyster Pond, MA, USA, August 2009 | [60] |
| N16961 | O1 El Tor | [61] |
| *E. coli* | | |
| β2163 | (F−) RP4-2-Tc::Mu Δ*dapA*::(*erm-pir*) (Kn Em) | [40] |
| CAG18439 | MG1655 *lacZU118 lacI42*::Tn*10* (Tc) | [62] |
| BW25113 | F⁻ Δ(*araD-araB*)567, Δ*lacZ4787*(::*rrnB-3*), λ⁻, *rph-1*, Δ(*rhaD-rhaB*)568, *hsdR514* | [41] |
| EC100 | F⁻ *mcrA* Δ(*mrr-hsdRMS-mcrBC*) Φ80d*lacZΔM15* Δ*lacX74 recA1 endA1 araD139* Δ(*ara, leu*)7697 *galU galK* λ⁻ *rpsL* (Sm^R) *nupG* | Epicentre, Madison Wis. |
| KH40 | MG1655 Δ*dapA*::*cat* (Cm) | This study |
| VB112 | Rf-derivative of MG1655 | [63] |
| GG56 | Nx-derivative of BW25113 | [35,64] |
| FD034 | CAG18439 Δ*lacZ dusA*'-'*lacZ-aad7* (Tc Sp) | This study |
| FD036 | GG56 *dusA*'-'*lacZ-aad7* (Nx Sp) | This study |
| Plasmids | | |
| pKD3 | Cm^R PCR template for one-step chromosomal gene inactivation (Cm) | [41] |
| pKD4 | Kn^R PCR template for one-step chromosomal gene inactivation (Kn) | [41] |
| pKD13 | Kn^R PCR template for one-step chromosomal gene inactivation (Kn) | [41] |
| pVI36 | Sp^R PCR template for one-step chromosomal gene inactivation (Sp) | [63] |
| pVI42B | pVI36 BamHI::*P_{lac}–lacZ* (Sp) | [65] |
| pSW23T | pSW23::*oriT*_{RP4}; *oriV*_{R6Kγ} (Cm) | [40] |
| pOP*lacZ* | pAH56 *lacZ* (Kn) | [17] |
| pBAD30 | *ori*_{p15A} *bla araC P*_{BAD} (Ap) | [66] |
| pBAD-*acaDC* | pBAD30::*acaDC* (Ap) | [17] |
| pBAD-*rdfN* | pBAD30::*rdfN* (Ap) | This study |
| pBAD-*sgaC* | pBAD30::*sgaC* (Ap) | This study |
| pVCR94^Kn Δ*acr2* | Δ*acr2* mutant of pVCR94^Kn (Su Kn) | [34] |
| pVCR94^Sp | Sp^R derivative of pVCR94 (Su Sp) | [17] |
| pVCR94^Sp Δ*acaC* | Δ*acaC* mutant of pVCR94^Sp (Su Sp) | [17] |
| pVCR94^Sp Δ*acaDC* | Δ*acaDC* mutant of pVCR94^Sp (Su Sp) | [17] |
| Integrative elements | | |
| IME*Vch*USA3 | | This study |
| IME*Vch*USA3^Cm | IME*Vch*USA3 CGT85_RS05425Ω pSW23T (Cm) | This study |
| IME*Vch*USA3^Kn | Kn^R derivative of IME*Vch*USA3 (Kn) | This study |
| IME*Vch*USA3^Kn Δ*sgaC* | Δ*sgaC* mutant of IME*Vch*USA3 (Kn) | This study |
| IME*Vch*USA3^Kn Δ*rdfN* | Δ*rdfN* mutant of IME*Vch*USA3 (Kn) | This study |
| mini-IE | *attP-int-aad7* derived from IME*Vch*USA3 (Sp) | This study |

[a]Ap, ampicillin; Cm, chloramphenicol; Em, erythromycin; Kn, kanamycin; Nx, Nalidixic acid; Rf, rifampin; Sm, streptomycin; Sp, spectinomycin; Su, sulfamethoxazole; Tc, tetracycline; Tm, trimethoprim; ts, thermosensitive.

plasmid pSW23T [40] at locus CGT85_RS05425 of *V. cholerae* OYP6G08 (Genbank NZ_NMSY01000009) by homologous recombination. Briefly, CGT85_RS05425 was amplified using primer pair dusAigEcoRIF/dusAigEcoRIR. The amplicon was digested with EcoRI and cloned into EcoRI-digested pSW23T using T4 DNA ligase. The resulting plasmid was confirmed by restriction profiling and DNA sequencing, then introduced into the DAP-

auxotrophic *E. coli* β2163 [40] by transformation. This strain was used as a donor in a mating assay to transfer the plasmid into *V. cholerae* OYP6G08, generating IME*Vch*USA3$^{Cm}$. Single-copy integration of the pSW23T derivative was confirmed by PCR and antibiotic resistance profiling.

IME*Vch*USA3$^{Kn}$ was constructed from IME*Vch*USA3$^{Cm}$. Briefly, pVCR94$^{Kn}$ Δ*acr2* was transferred from the DAP-auxotrophic *E. coli* KH40 into OYP6G08 bearing IME*Vch*USA3$^{Cm}$. After selection on LB agar medium supplemented with chloramphenicol and kanamycin, Cm$^{R}$ Kn$^{R}$ *V. cholerae* OYP6G08 transconjugants were confirmed by growth on thiosulfate-citrate-bile salts-sucrose (TCBS) agar medium (Difco). In *V. cholerae*, the integration and excision of the IME were confirmed by amplification of the *attL*, *attR*, *attB*, and *attP* sites with primer pairs oRD4/ORD6, oRD1/oRD3, oRD1/oRD6, and oRD4/oRD3, respectively. IME*Vch*U-SA3$^{Cm}$ was then mobilized from OYP6G08 to *E. coli* CAG18439. In *E. coli*, the integration and excision of the IME were confirmed by amplification of the *attL*, *attR*, *attB* and *attP* sites with primer pairs oRD4/ORD5, oRD2/oRD3, oRD2/oRD5 and oRD4/oRD3, respectively. IME*Vch*USA3$^{Kn}$ was constructed by replacing pSW23T with a single kanamycin resistance marker using the one-step chromosomal gene inactivation technique with primer pair dusAscar-NoFRTf/dusAscarNoFRTr and pKD13 as the template. The deletions Δ*sgaC* and Δ*prtN* in IME*Vch*USA3$^{Kn}$ were obtained using the primer pairs oFD26r/oFD26f and DelprtNr/DelprtNf, and pKD3 and pVI36 as the templates, respectively. The Δ*dapA* deletion mutant of *E. coli* MG1655 was constructed using primer pair FwDeltaDapA-MG1655/ RvDeltaDa-pA-MG1655 and pKD3 as the template. The Δ*lacZ* mutation was introduced in *E. coli* CAG18439 using primer pair lacZW-B/lacZW-F and plasmid pKD4 as the template. The *dusA'-'lacZ* fusion was introduced in *E. coli* BW25113 and CAG18439 using primer pair oDF15/oDF16 and pVI42B as the template. The fortieth codon of *dusA* was fused to the eighth codon of *lacZ* downstream of the *attB* site. The λRed recombination system was expressed using either pSIM6, pSIM9 or pKD46 [41,42]. When appropriate, resistance cassettes were excised from the resulting constructions using the Flp-encoding plasmid pCP20 [43]. All deletions were validated by antibiotic profiling and PCR.

Fragments encompassing promoter regions upstream of *int*, *traN*, *traG*, *s018* and *rdfN* were amplified using primer pairs oFD6.f/oFD6.r, oFD1.f/oFD1.r, oFD3.f/oFD3.r, oFD5.f/oFD5.r and oFD4.f/oFD4.r, respectively, and genomic DNA from *E. coli* CAG18439 *dusA*::IME*Vch*U-SA3$^{Kn}$ as the template. The amplicons were digested with PstI/XhoI and cloned into PstI/XhoI-digested pOP*lacZ* [17]. The resulting constructs were single-copy integrated into the *attB*$_{λ}$ chromosomal site of *E. coli* BW25113 using pINT-ts [44]. To construct the expression vectors pBAD-*rdfN* and pBAD-*sgaC*, PCR fragments containing *rdfN* or *sgaC* were amplified from genomic DNA of *E. coli* CAG18439 bearing IME*Vch*USA3 as the template and primer pairs prtNEcoRIf/prtNHindIIIrev and oFD38r/oFD44f, respectively. The PCR fragments were digested by either EcoRI or SacI, and HindIII and cloned into pBAD30 cut with the same enzymes.

mini-IE was constructed as follows. The 1,591-bp fragment of excised circular IME*Vch*U-SA3$^{Kn}$ that contains *attP-int* was amplified using primer pair oVB12/oVB10 and genomic DNA from *E. coli* CAG18439 *dusA*::IME*Vch*USA3$^{Kn}$ as the template. The 1,421-bp fragment of pVI36 that contains *aadA7* was amplified using primer pair oVB11/oVB13. Both fragments were joined using the PCR-based overlap extension method [45]. After the final PCR amplification using oVB12/oVB13, the amplicon was purified, digested with SacI, and ligated. The ligation mixture was then transformed into *E. coli* EC100. Transformant colonies were selected on LB agar supplemented with spectinomycin. The constitutive expression of *int* and the absence of replicon prompted the spontaneous integration of mini-IE at the 5' end of *dusA* in EC100.

All final constructs were verified by PCR and DNA sequencing by the Plateforme de Séquençage et de Génotypage du Centre de Recherche du CHUL (Québec, QC, Canada).

## qPCR assays

qPCR assays for quantification of excision and copy number of IME*Vch*USA3[Kn] were carried out as described previously [22] with the following modification. $attB_{dusA}$ (241 bp) and *higA* (229 bp) of IME*Vch*USA3[Kn] were quantified using primer pairs attBdusAqPCRfwd/ attBdusAqPCRrev and higAqPCRfwd/ higAqPCRrev, respectively (S2 Table). The excision rate and copy number of IME*Vch*USA3[Kn] were calculated as the ratio of free $attB_{dusA}$ site per chromosome and as the ratio of *higA* per chromosome, respectively. The data were analyzed and normalized using all three chromosomal genes *dnaB*, *hicB* and *trmE* as references and the qBase framework as described previously [22,46].

## β-galactosidase assays

The assays were carried out on LB agar plates supplemented with 5-bromo-4-chloro-3-indolyl-β-D-galactopyranoside (X-gal) or in LB broth using *o*-nitrophenyl-β-D-galactopyranoside (ONPG) as the substrate as described previously [32]. *acaDC* expression from pBAD-*acaDC* was induced by adding 0.2% arabinose to a refreshed culture grown to an $OD_{600}$ of 0.2, followed by a 2-h incubation at 37°C with shaking prior to cell sampling.

## Comparative analyses

Sequences were obtained using blastp against the Genbank Refseq database with the primary sequences of key proteins MpsA, TraG$_S$, SgaC, TraN$_S$ of SGI1 (Genbank AAK02039.1, AAK02037.1, AAK02036.1, AAK02035.1, respectively), and Int$_{dusA}$ of IE*Vch*Bra2 (Genbank EEO15317.1) and Int$_{yicC}$ of IE*Eco*MOD1 (Genbank WP_069140142.1). Hits were exported, then sorted by accession number to identify gene clusters that likely belong to complete IEs. Sequences of IEs were manually extracted and the extremities were identified by searching for the direct repeats contained in *attL* and *attR* sites. When an IE sequence spanned across two contigs (e.g., IE*Vch*Hai10 and IE*Ppl*Ind1), the sequence was manually assembled. IE sequences were clustered using cd-hit-est with a 0.95 nucleotide sequence identity cut-off [47]. Some of the annotated sequences were manually curated to correct missing small open reading frames such as *mpsB*, and inconsistent start codons. Pairwise comparisons of Int, MpsA, TraG, SgaC and TraN proteins were generated with blastp using sets of representative proteins selected after clustering using cd-hit with a 0.95 sequence identity cut-off (Int, MpsA, TraG, SgaC) or a 0.90 sequence identity cut-off (TraN) [47]. Heatmaps showing the blastp identity scores were drawn using the Python library seaborn v0.11.1 [48]. Circular blast representations (blast atlases) were generated with the Blast Ring Image Generator (BRIG) 0.95 [49], with blastn or blastp, against SGI1ΔIn104 and IE*Vch*USA2, with an upper identity threshold of 80% and a lower identity threshold of 60%. Antibiotic resistance gene prediction was conducted using the Resistance Gene Identifier (RGI) software and CARD 3.1.3 database [50]. AcaCD binding motifs were identified using FIMO and MAST [51] with the AcaCD motif matrix (S1 Matrix) described previously [17]. Logos for *attL* and *attR* repeats were generated with MAST [51] using alignments of sequences flanking the IE$_{dusA}$ elements identified in this work.

## Phylogenetic analyses

Evolutionary analyses were conducted in MEGA X [52] and inferred by using the maximum likelihood method based on the JTT (MpsA or SgaC proteins), LG (Int$_{dusA}$, Int$_{yicC}$, TraG or

RepA$_{IncFII}$ proteins) or WAG (TraN) matrix-based models [53–55]. Protein sequences were aligned with Muscle [56]. Aligned sequences were trimmed using trimal v1.2 using the automated heuristic approach [57]. Initial tree(s) for the heuristic search were obtained automatically by applying Neighbor-Join and BioNJ algorithms to a matrix of pairwise distances estimated using a JTT model, and then selecting the topology with the superior log likelihood value. A discrete Gamma distribution was used to model evolutionary rate differences among sites (5 categories) for Int$_{dusA}$ (parameter = 3.5633), Int$_{yicC}$ (parameter = 2.6652), SgaC (parameter = 1.4064), TraG (parameter = 1.9005) and TraN (parameter = 1.6476) proteins. For Int$_{dusA}$, MpsA and TraG, the rate variation model allowed for some sites to be evolutionarily invariable ([+I], 7.81% sites for Int$_{dusA}$, 44.62% sites for MpsA and 5.22% sites for TraG$_S$). The trees are all drawn to scale, with branch lengths measured in the number of substitutions per site. In all trees, bootstrap supports are shown as percentages at the branching points only when > 80%.

*oriT* sequences were obtained manually using the previously identified *oriT* of SGI1 as the reference [19], then clustered using cd-hit-est with a 1.0 nucleotide sequence identity cut-off. Sequences were then aligned using Muscle and a NeighborNet phylogenetic network was built using SplitsTree4 [58] with default parameters (Uncorrected_P method for distances and EqualAngle drawing method). The secondary structures of the aligned *oriT* sequences were predicted using RNAalifold v2.4.17 from the ViennaRNA package [59]. Default options were used (including no RIBOSUM scoring), except for the following: no substituting "T" for "U" (—noconv), no lonely pairs (—noLP), no GU pairs (—noGU) and DNA parameters (-P DNA). The predicted Vienna output and the annotated alignment were merged into a predicted secondary structure of SGI1 *oriT* color-coded to display the inter-island diversity.

## Statistical analyses and figures preparation

Numerical data presented in graphs are available in S3 Dataset. Prism 8 (GraphPad Software) was used to plot graphics and to carry out statistical analyses. All figures were prepared using Inkscape 1.0 (https://inkscape.org/).

## Supporting information

**S1 Fig. Comparative sequence analysis of SGI1-like *dusA*-specific IEs.** Blastn and blastp atlases using either SGI1ΔIn104 (A) or IE*Vch*USA2 (B) as the reference. Coding sequences appear on the outermost circle in blue for the positive strand and red for the negative strand, with the *oriT* depicted as a grey arc. All other sequences are represented only according to their homology with the reference, with full opacity corresponding to 100% identity and gaps indicating identity below 60%. The order of the IEs in the atlases is indicated according to the color keys shown in the inset of panel B.
(PDF)

**S2 Fig. NeighborNet phylogenetic network (A) and predicted secondary structure of 39 *oriT* loci (B) of SGI1-like IEs.** Each IE's integration site and type are annotated. The sequence of canonical SGI1 (Genbank AF261825.2) was used as a reference to show the predicted secondary structure of all *oriT* sequences. Pairs can be perfectly conserved, imperfectly conserved (1/39 not conserved), not conserved (> 1/39), or an A-T or G-C pair only. In the latter case, the sequence is not conserved, but the predicted local secondary structure is.
(PDF)

**S3 Fig. Maximum likelihood phylogenetic analysis of key proteins of SGI1-related IEs.** The trees for MpsA (A), TraG (B), SgaC (C) and TraN (D) proteins are drawn to scale, with branch

lengths measured in the number of substitutions per site over 321, 1,145, 188, and 968 amino acid positions, respectively. For clarity, the lengths of the branches linking the two groups in panels A and C were artificially divided by 8 and 4, respectively. Taxa corresponding to IEs targeting *trmE* and *yicC* are shown by a light blue circle and a red circle, respectively. All other taxa correspond to *dusA*-specific IEs. Proteins accession numbers are provided in S1 Table and S2 Dataset.
(PDF)

**S4 Fig. Alignment of AcaCD-responsive promoters predicted in IEs targeting *dusA*, *yicC* and *trmE*.** Promoter sequences are grouped based on the function of the expressed genes as follows: (A) RDFs; (B) mating pair stabilization; (C) mating pair formation and stabilization; (D) unknown. AcaCD binding sites are shown in green. Logo sequences and *p*-values were generated by MAST [51]. Known transcription start sites are shown in blue [17,22]. Predicted Shine-Dalgarno sequences are shown in pink. The initiation start codon is shown in bold letters.
(PDF)

**S5 Fig. Excision of IME*Vch*USA3 is enhanced in IncC⁺ cells.** (A) Model of excision of IME*Vch*USA3. (B and C) Detection of *attB*, *attP*, *attL* and *attR* sites by PCR in colonies of *V. cholerae* OYO6G08 bearing (lanes 9 to 16) or lacking (lanes 1 to 8) pVCR94^Kn Δ*acr2*. Control lanes: L, 1Kb Plus DNA ladder (Transgen Biotech); +, *V. cholerae* N16961 genomic DNA. (D) Detection of *attB*, *attP*, *attL* and *attR* sites by PCR in transconjugant colonies of *E. coli* CAG18439 (lanes 1 to 4). L, 100bp Plus II DNA Ladder (Transgen Biotech)
(PDF)

**S1 Table. Features of the identified IEs and associated strains.**
(XLSX)

**S2 Table. Oligonucleotides used in this study.**
(DOCX)

**S1 Dataset. Features of ORFs in the identified IEs.**
(XLSX)

**S2 Dataset. Clusters generated by cd-hit for Int, MpsA, TraG, SgaC, and TraN.**
(XLSX)

**S3 Dataset. Numerical data presented in Figs 4–6.**
(XLSX)

**S1 Matrix. AcaCD motif matrix to identify AcaCD binding sites.**
(TXT)

## Acknowledgments

We are grateful to Yann Boucher for the kind gift of *Vibrio cholerae* OYP6G08 and Kévin T. Huguet for technical assistance. We thank Nicolas Rivard and David Roy for their insightful comments on the manuscript.

## Author Contributions

**Conceptualization:** Romain Durand, Vincent Burrus.

**Data curation:** Vincent Burrus.

**Formal analysis:** Romain Durand, Florence Deschênes, Vincent Burrus.

**Funding acquisition:** Vincent Burrus.

**Investigation:** Romain Durand, Florence Deschênes, Vincent Burrus.

**Methodology:** Romain Durand, Florence Deschênes, Vincent Burrus.

**Project administration:** Vincent Burrus.

**Resources:** Vincent Burrus.

**Supervision:** Romain Durand, Vincent Burrus.

**Validation:** Romain Durand, Florence Deschênes, Vincent Burrus.

**Visualization:** Romain Durand, Florence Deschênes, Vincent Burrus.

**Writing – original draft:** Romain Durand, Vincent Burrus.

**Writing – review & editing:** Romain Durand, Florence Deschênes, Vincent Burrus.

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
