## [Decision Letter · Decision Letter 0]

13 Jul 2021

Dear Dr Burrus,

Thank you very much for submitting your Research Article entitled 'Genomic islands targeting dusA in Vibrio species are distantly related to Salmonella Genomic Island 1 and mobilizable by IncC conjugative plasmids' to PLOS Genetics.

The manuscript was fully evaluated at the editorial level and by independent peer reviewers. Unfortunately we were unable to obtain more that two reviews and in the interest of not holding up your work for longer than necessary we have decided to proceed. The two reviewers had significantly divergent opinions on the merits of your work. The reviewers appreciated the attention to an important problem, but raised some substantial concerns about the current manuscript. Based on the reviews, we will not be able to accept this version of the manuscript, but we would be willing to review a much-revised version. We cannot, of course, promise publication at that time.

Should you decide to revise the manuscript for further consideration here, your revisions should address the specific points made by each reviewer. We strongly encourage you to address all of the comments and criticisms fully. We will also require a detailed list of your responses to the review comments and a description of the changes you have made in the manuscript.

If you decide to revise the manuscript for further consideration at PLOS Genetics, please aim to resubmit within the next 60 days, unless it will take extra time to address the concerns of the reviewers, in which case we would appreciate an expected resubmission date by email to plosgenetics@plos.org.

[LINK]

We are sorry that we cannot be more positive about your manuscript at this stage. Please do not hesitate to contact us if you have any concerns or questions.

Yours sincerely,

Diarmaid Hughes

Associate Editor

PLOS Genetics

Josep Casadesús

Section Editor: Prokaryotic Genetics

PLOS Genetics

Reviewer's Responses to Questions

**Comments to the Authors:**

Reviewer #1: This manuscript describes novel genomic islands related to SGI1, identified in other target sites and other bacterial species than reported for SGI1 since 20 years. The islands were functionally assessed for their mobility in relation with IncC plasmids required for their mobilization. The authors propose an interesting evolutionary model based on the data of this study that may have resulted in the epidemic SGI1 and its variants. The manuscript is clearly written and I have only minor comments or questions:

1. The authors have focused their study on mobilization genes of the new islands identified. Do these islands carry genes of medical importance such as antimicrobial resistance genes or other genes that may confer an advantage to the bacterial pathogen such as virulence genes? If so the authors may consider including this information in the text.

2. For SGI1 besides trmE another integration site has been reported (Doublet et al., PLoS One. 2008 Apr 30;3(4):e2060. doi: 10.1371/journal.pone.0002060). In some genomes SGI1 relatives occur in this integration site.

3. Lines 75 and 117, SGI1 is presented as one of the most "atypical" and most studied IMEs. The use of "atypical" in these lines is not clear to me. Why is SGI1 atypical?

Reviewer #2: Overall this is potentially a solid study that is of interest because it extends the group of IME that use the unique IncC plasmid-integrative mobilizable element (IME) relationship studied to date mainly in the case of the IME SGI1 and variants of SGI1. However, there are problems with presentation - it’s jumbled - that does not help readers follow the stories presented. Thinking through the findings of the work to consider what to include, to find a more logical order, and improvement of the Figures would be very beneficial.

This study identifies a series of integrative mobilizable elements (IME) that include genes that are homologues of some of the important genes found in SGI1 (and relatives), the best studied of the bacterial IME. In SGI1, these genes are associated with mobility and dependence on an IncA or IncC plasmid for intercellular transfer. The novel IME, mostly found in Vibrio genomes, are not found in the same location as SGI1 because they encode different Int to that used by SGI1 and hence integrate at 2 different locations, dus for the Vibrio set and yicC for the others. This is not particularly surprising as groups of Int are known to be specific for a specific site and the sites are ones known to be recognised by specific groups of Int.

This aspect of the work is not well analysed or well organised or presented. The analysis is minimal and does not include critical information that is standard for this type of work such as accession numbers for the sequences used in phylogenies - they can be listed in the Fig legend. The Figures are not of publication quality and Fig. 2 is not the best to complement or inform the text.

The study also addresses some initial basic questions relating to the properties of one of the IME (called MGIVchUSA3) that targets dus, including showing that:

1. The IME is excised in the presence of an IncC plasmid and this is dependent on rdf, thus identifying the gene for the recombination directionality factor (rdf).

2. Only 2 of 4 potential AcaCD binding sites identified are AcaDC responsive – one is upstream of rdf. So, this IME may not have been the best choice to study.

3. SgaC, the AcaC homologue encoded by the IME, can substitute for AcaC (interacting successfully with AcaD) as there is no sgaD gene in this IME

4. IncC plasmid-assisted transfer.

5. IME that use the 5’-end of dus supply a replacement 5’-end and a promoter.

However, a well annotated GenBank entry (a TPA is possible if the sequence belongs to others) for the IME used is needed to follow this work but I could not find reference to one. This MUST be supplied.

Finally, a highly speculative model for the evolution of the IncC-mobilizable group is presented.

Major comments:

1. Use of MGI and GI. Why call an IME MGI? Why use GI? GI is a very generic term that be used for both IE and Tn as well as diverged regions replaced by homologous recombination. IE must integrate using a site-specific recombinase. In 2021, it’s time to be more accurate. Also, Kn for kanamycin! There are 2 accepted abbreviations Km or KAN. One of these should be used.

2. The Introduction could be a bit more compact.

3. There are too many references (73 currently). Those that are not directly relevant are not needed, e.g. ref. 8, this information is reviewed in refs. 12, 13 and 14, and should be culled extensively.

4. Figures are not of publication quality, particularly Figs. 1 and 2 which are blurred and have font size verging on unreadable. In Figure 1, the various blue shades are not easily distinguished.

5. Splitting Fig. 1 to show more clearly the features of SGI1 (for the Introduction) and the IME used experimentally (for the Results) would be helpful. The yicC group could also be separated.

6. Re-organisation of the components of other Figures to better group items related to the same issue and put them in an order more consistent with the text is needed.

7. Fig. legends should describe the Figure not the Methods used. Please fix where relevant.

8. lines 106-7, Other IE, particularly ICE, do carry rep genes and replicate.

9. line 132, Is “resembling” the right word here. In the Introduction it’s said they don’t have to resemble. Perhaps “sharing features with”.

10. A simple Table showing the basic features of the IME shown in Fig. 1, e.g size, organism, and where their sequences can be found (Accession numbers) is needed in the main body of the article.

11. Please state the consensus used to predict AcaCD binding sites. The legend to Figure 1 must clearly state that the marked AcaCD binding sites are potential; you later show 2 are not responsive.

12. what % identity is indicated by the grey shading in Fig. 1; is it identity of DNA or proteins. Please add this information to the legend.

13. Some % identities (or ranges) for the similar proteins are needed in the text.

14. Line 115, “unrelated” Surely not, they are both tyrosine recombinases.

15. Line 165, which 5? Please state here or explain how to find them in the Fig.

16.lines 176-189, one name (GIVchBan1) for 2 IE – that’s not sensible. It’s well known that a further IE can integrate where one is already present. So, each should have a name.

17. lines 182-233. This section is close to impossible to follow. It needs a vast improvement if any reader is to follow it. Which genes are present is in Fig..1. So, the BRIG alignments in Figure 2 are not helpful (may be OK as supplementary). Some detail as the to % identities of genes/proteins within specific groups is essential here. This can easily be tabulated and this Table is needed in this section. Only the phylogeny for MpsA-TraG-SgaC-TraN (and the supplementary phylogenies for individual genes in Fig. S2) are needed here. The Int proteins were discussed in the previous section.

18. VchN2708 is the outlier to type 3 in 2A but Vch2719 is the only outlier in S2. Are they labelled correctly?

19. Lines 266-293, Why was this done with a de-repressed IncC plasmid? This section is not needed and should be removed. The information and Figure could be Supplementary.

20. In Figs., 5A and 6D, why is attR on the left and attL on the right? This is confusing. In addition, usually L is near the int gene. It’s easy to fix.

Please convert the bar graphs to scatter plots so that the number and range of values determined can be seen.

21. Line 304, so the IME transfer frequency is higher when assisted by the standard plasmid compared to with the de-repressed equivalent? Please explain.

22. I’ll stop here and say, a manuscript should be better prepared before submission. It is not a reviewer’s job to do this much work.

Minor comments:

Lines 86-7, add florfenicol.

Line 97, Isn’t the HKL group defined in ref. 14? “Seem to”? There are sequences.

Line 101, add the hyphen to 3’ end here and elsewhere and 5’ end elsewhere.

What is meant be the text of lines 118-120. Please clarify. Are you simply saying you used protein sequences in your searches as is quite standard?

Line 194-5, do the authors of ref. 25 really call MspA a tyrosine recombinase or just allude to similarities?

**Have all data underlying the figures and results presented in the manuscript been provided?**

Reviewer #1: Yes

Reviewer #2: **No: **Transfer frequency data is not clear because bar graphs rather than scetter plots have been used.

PLOS authors have the option to publish the peer review history of their article (what does this mean?). If published, this will include your full peer review and any attached files.

Reviewer #1: No

Reviewer #2: No

---

## [Editor Report · Decision Letter 1]

9 Aug 2021

Dear Dr Burrus,

We are pleased to inform you that your manuscript entitled "Genomic islands targeting dusA in Vibrio species are distantly related to Salmonella Genomic Island 1 and mobilizable by IncC conjugative plasmids" has been editorially accepted for publication in PLOS Genetics. Congratulations!

We, the two editors, have carefully read your response letter and your revised manuscript, and we appreciate your honesty and attention to detail in revising this manuscript in line with the comments of the two reviewers. We have decided to accept your revised manuscript without submitting it to a further round of review.

Yours sincerely,

Diarmaid Hughes

Associate Editor

PLOS Genetics

Josep Casadesús

Section Editor: Prokaryotic Genetics

PLOS Genetics

Comments from the reviewers (if applicable):

**Data Deposition**

http://datadryad.org/submit?journalID=pgenetics&manu=PGENETICS-D-21-00821R1

**Press Queries**

---

## [Editor Report · Acceptance letter]

17 Aug 2021

PGENETICS-D-21-00821R1 

Genomic islands targeting *dusA* in *Vibrio* species are distantly related to *Salmonella* Genomic Island 1 and mobilizable by IncC conjugative plasmids 

Dear Dr Burrus, 

We are pleased to inform you that your manuscript entitled "Genomic islands targeting *dusA* in *Vibrio* species are distantly related to *Salmonella* Genomic Island 1 and mobilizable by IncC conjugative plasmids" has been formally accepted for publication in PLOS Genetics! Your manuscript is now with our production department and you will be notified of the publication date in due course.

With kind regards,

Agnes Pap

PLOS Genetics

On behalf of:
